# Structural and Functional Properties of Kappa Tropomyosin

**DOI:** 10.3390/ijms24098340

**Published:** 2023-05-06

**Authors:** Galina V. Kopylova, Anastasia M. Kochurova, Daria S. Yampolskaya, Victoria V. Nefedova, Andrey K. Tsaturyan, Natalia A. Koubassova, Sergey Y. Kleymenov, Dmitrii I. Levitsky, Sergey Y. Bershitsky, Alexander M. Matyushenko, Daniil V. Shchepkin

**Affiliations:** 1Institute of Immunology and Physiology, Russian Academy of Sciences, 620049 Yekaterinburg, Russia; g_rodionova@mail.ru (G.V.K.); kochurova.a.m@mail.ru (A.M.K.); cmybp@mail.ru (D.V.S.); 2Research Center of Biotechnology, Russian Academy of Sciences, 119071 Moscow, Russia; daria_logvinova@mail.ru (D.S.Y.); victoria.v.nefedova@mail.ru (V.V.N.); s.yu.kleymenov@gmail.com (S.Y.K.); levitsky@inbi.ras.ru (D.I.L.); ammatyushenko@mail.ru (A.M.M.); 3Institute of Mechanics, Moscow State University, 119192 Moscow, Russia; andrey.tsaturyan@gmail.com (A.K.T.); nkoubassova@gmail.com (N.A.K.); 4Koltzov Institute of Developmental Biology, Russian Academy of Sciences, 119334 Moscow, Russia

**Keywords:** cardiac tropomyosin isoforms, thin filament, atrial and ventricular myosin, actin–myosin interaction, in vitro motility assay

## Abstract

In the myocardium, the *TPM1* gene expresses two isoforms of tropomyosin (Tpm), alpha (αTpm; Tpm 1.1) and kappa (κTpm; Tpm 1.2). κTpm is the result of alternative splicing of the *TPM1* gene. We studied the structural features of κTpm and its regulatory function in the atrial and ventricular myocardium using an in vitro motility assay. We tested the possibility of Tpm heterodimer formation from α- and κ-chains. Our result shows that the formation of ακTpm heterodimer is thermodynamically favorable, and in the myocardium, κTpm most likely exists as ακTpm heterodimer. Using circular dichroism, we compared the thermal unfolding of ααTpm, ακTpm, and κκTpm. κκTpm had the lowest stability, while the ακTpm was more stable than ααTpm. The differential scanning calorimetry results indicated that the thermal stability of the N-terminal part of κκTpm is much lower than that of ααTpm. The affinity of ααTpm and κκTpm to F-actin did not differ, and ακTpm interacted with F-actin significantly worse. The troponin T1 fragment enhanced the κκTpm and ακTpm affinity to F-actin. κκTpm differently affected the calcium regulation of the interaction of pig and rat ventricular myosin with the thin filament. With rat myosin, calcium sensitivity of thin filaments containing κκTpm was significantly lower than that with ααTpm and with pig myosin, and the sensitivity did not differ. Thin filaments containing κκTpm and ακTpm were better activated by pig atrial myosin than those containing ααTpm.

## 1. Introduction

Alpha-tropomyosin (αTpm; Tpm 1.1), a product of the *TPM1* gene, is essential for heart development and supporting its contractile function. Abnormalities of expression of the αTpm lead to myofibrillogenesis violations, defects of myocardium development, and embryos’ death at E9.5, as shown in the mouse model [1,2]. The impairment of myofibrillogenesis may occur due to stabilizing influence of tropomyosin on the actin filament in the structure of the thin filament.

In addition to αTpm, there are two more isoforms of tropomyosin in the mammalian myocardium: beta-tropomyosin (βTpm; Tpm 2.2), a product of the *TPM2* gene [3], and kappa-tropomyosin (κTpm; Tpm 1.2), which is the result of alternative splicing of the *TPM1* gene [4]. In the mRNA encoding κTpm, exon 2b (striated muscle tissue and non-muscle Tpm isoforms) is replaced by exon 2a (smooth muscle tissue) that corresponds to about 40 amino acid residues (from 39 to 77 a.a.) in the tropomyosin molecule [4,5].

The proportion of the Tpm β-chain in the heart of mammals varies depending on the size of the animal [3,6]. In the heart of the adult mice, it is negligible (almost absent), while in the sheep and bovine myocardium, it is up to 15–20% [3]. In striated muscle tissue, the Tpm β-chain forms a heterodimer with the Tpm α-chain [3]. According to recent data, the human myocardium contains about 5% Tpm β-chain [6,7,8], while κTpm in the myocardium of mammals and humans is about 3–4% [7]. Mass spectrometry showed that more κTpm is expressed in the atria than in the ventricles [8]. κTpm expression is upregulated in heart failure and dilated cardiomyopathy [7].

αTpm and κTpm differ structurally and functionally. In transgenic mice, increased expression of κTpm leads to systolic and diastolic dysfunction and dilated phenotype of the heart chamber [7,9]. Rajan et al. [7] showed that κTpm is less thermally stable, but no detailed studies of its structure have been carried out. It was also found that κTpm has a lower affinity to F-actin than αTpm [7]. In the ventricular myocardium of transgenic mice expressing κTpm, dilated cardiomyopathy developed without fibrosis formation and cardiomyocyte structure disordering [7,9]. The presence of κTpm led to a decrease in the calcium sensitivity tension relationship of skinned myocardial preparations but did not affect the maximum tension [7,9]. On the Mexican axolotl model, suppression of the κTpm expression in the developing heart impaired myofibrillogenesis in cardiomyocytes and heart development [4,10]. Note that the presence of a minor amount of κTpm in the adult axolotl myocardium was also found [4,10].

Thus, even though the κTpm amount is small, its role in myocardium development and its contractile function can be perceptible. However, the role of κTpm in regulating actin–myosin interaction in the myocardium has never been investigated. We studied the structural features of κTpm and its regulatory function in the atrial and ventricular myocardium using the in vitro motility assay. In addition, we tested the possibility of Tpm heterodimer formation from α- and κ-chains.

## 2. Results

### 2.1. Properties of Coiled-Coil Tropomyosin Dimers

#### 2.1.1. Formation of ακ-Heterodimers of Tpm

To evaluate the ability of αTpm and κTpm chains to form a heterodimer, ααTpm, κκTpm, or their equimolar mixture were heated to 70 °C to dissociate the dimers. After heating, the solutions were cooled, and Tpm dimers formed. Then, Tpm dimers were cross-linked at SH groups using DTNB as described earlier [11]. In the case of an equimolar ααTpm and κκTpm mixture, three types of dimers could be formed: ααTpm or κκTpm homodimers or ακTpm heterodimers. The mixture composition was analyzed by SDS-PAGE in the absence of β-mercaptoethanol. The formation of the ακTpm heterodimers was thermodynamically favorable (Figure 1; lane 5, −βM). The predominant optical density in lane 5 was located between the positions of αTpm (lane 4 −βM) and κTpm (lane 6 −βM) bands.

#### 2.1.2. Formation of ακ-Heterodimers of Tpm

We applied circular dichroism (CD) to compare the thermal unfolding of ααTpm, ακTpm, and κκTpm. The structure of ααTpm, ακTpm, and κκTpm corresponds to the typical structure of the α-helix but the thermal stability of these Tpm isoforms was substantially different (Figure 2, Table 1). The κκTpm isoform had the lowest stability while the ακTpm heterodimer was even more stable than the ααTpm homodimer (Figure 2b). All melting curves had three transitions except for the curve for κκTpm. An increase in the stability of ακTpm may be related to the formation of more bonds between α and κ chains in the heterodimer compared to homodimer Tpm molecules, which consist of only one type of chain.

#### 2.1.3. Thermal Unfolding and Domain Structure of κκTpm Studied with Differential Scanning Calorimetry

We applied differential scanning calorimetry (DSC) to investigate the thermal unfolding and domain structure of κκTpm. DSC experiments were performed under conditions when all Tpm isoforms were in an entirely reduced state (see Section 4 for more details).

The results of the deconvolution of DSC curves for ααTpm and κκTpm into individual thermal transitions (calorimetric domains) are presented in Figure 3, and the main calorimetric parameters for these domains are summarized in Table 2. 

ααTpm demonstrated three calorimetric domains on the DSC curve (Figure 3), which were identified in the previous works [13,14,15]. Two of them, domains 2 and 3, can be assigned to the thermal unfolding of the C-terminal and N-terminal parts of the Tpm molecule, respectively [13,15]. As for the least thermostable domain 1, it was supposed to correspond to the melting of other parts of the molecule with reduced thermal stability, such as its middle part [15,16] or the head-to-tail overlap junction between the N- and C-termini of neighbor Tpm molecules [14].

The thermal unfolding of κκTpm was different from that of ααTpm (Figure 3). In particular, the calorimetric domains 2 and 3 of κκTpm were much less thermostable, by 5–8 °C, than these domains in ααTpm (Figure 3, Table 2). This observation prompted us to identify these domains in the κκTpm molecule, i.e., to reveal their correspondence to certain parts of the molecule. For this purpose, we performed cross-linking of κκTpm by the formation of a disulfide bond between Cys residues in two chains of the κκTpm dimer. The cross-linking was previously shown to significantly increase the thermal stability of the Tpm region where Cys residues are located [13,15]. The results of these experiments were performed by the κκTpm heating in the absence of reducing agents.

The DSC curve obtained from the first heating of the κκTpm sample (Figure 4) did not significantly differ from that for κκTpm in its reduced state (Figure 3), thus indicating that inter-chain disulfide cross-linking did not occur yet under these conditions. However, the DSC profile obtained from the second heating of the sample (Figure 4) showed significant changes in the thermal unfolding and domain structure of κκTpm. These changes were expressed in the appearance of the new highly thermostable calorimetric domain on the DSC thermogram (domain 4 at 52.4 °C) and the corresponding significant decrease in the enthalpy of domain 3 (Figure 4 and Table 3). Upon subsequent heatings, the enthalpy of domain 4 increased while the enthalpy of domain 3 decreased (Table 3). These results indicated that the calorimetric domain 3, whose stability increased due to disulfide cross-linking upon heating, reflects the thermal unfolding of those regions of the κκTpm molecule, which contain Cys residues. The κκTpm isoform, like ααTpm, contains the only Cys residue, Cys190, in the C-terminal half of each chain of the Tpm dimer. Thus, one can conclude that the calorimetric domain 3 of κκTpm reflects the thermal unfolding of the C-terminal part of the molecule. Correspondingly, another main domain of κκTpm, calorimetric domain 2 of reduced κκTpm (Figure 3), can be assigned to the thermal unfolding of the N-terminal part of the molecule.

Comparing the DSC data for ααTpm and κκTpm isoforms in their reduced state (Figure 3 and Table 2), one can conclude that the thermal stability of the N-terminal half (domain 3 in ααTpm and domain 2 in κκTpm) is much less in κκTpm than in ααTpm. This destabilization of the N-terminal part of κκTpm molecule can easily be explained by the difference between ααTpm and κκTpm isoforms in the N-terminal exons of TPM1 gene: exon 2a for κκTpm and exon 2b for ααTpm [5].

It should be noted that thermal transitions of calorimetric domains 1 and 2 could not be separated on the DSC thermograms of cross-linked κκTpm (Figure 4) where both of these domains melted together into a new domain 2 with highly increased enthalpy (Table 3). It seems likely that this effect was caused by some increase in the thermal stability of domain 1 (e.g., due to changes in the thermal unfolding of the middle part of κκTpm molecule induced by cross-linking). As a result, this domain either melts together with the N-terminal part (calorimetric domain 2), or its thermal transition coincides with that of the N-terminal domain 2, and the two transitions cannot be separated on the DSC profile by deconvolution procedure.

The DSC results indicate that the thermal stability of the N-terminal part of κκTpm is much lower than that of ααTpm.

#### 2.1.4. Interaction of N- and C-Ends of ααTpm and κκTpm

The interaction between the N- and C-ends of the Tpm molecules was investigated by measuring the viscosity of the tropomyosin solution (see Section 4). It was found that the viscosity of the ααTpm solution (0.41 ± 0.04 mPa·s) is higher than that of κκTpm (0.09 ± 0.01 mPa·s), indicating a stronger interaction between the ends of the ααTpm molecules.

#### 2.1.5. Molecular Dynamics (MD) Simulation

To compare the characteristics of α- and κ-Tpm isoforms in silico, we performed the MD simulations using the GROMACS v. 2019.3 software package (see Section 4). The stability of the α-helical structure of the Tpm isoforms was estimated from the average occupancy of the backbone hydrogen bonds (*h*-bonds) between the *n* and *n* + 4 residues in both chains. At the temperature of 300 K, the distribution of the *h*-bond occupancy along the molecule was similar for the three isoforms with the least stable regions at around the two intrinsically unstable areas around the non-canonical residues 137 and 218 [17,18] (Figure 5a). No significant changes were observed when the MD simulations were performed at the temperature raised to 315 K. At 323 K both isoforms were less stable. The ααTpm lost α-helical structure in the middle part around residue 137, and all three isoforms destabilized near residue 218 (Figure 5b). Surprisingly not much difference in the *h*-bonds occupancy was observed in the N-terminal region of Tpm residues 40–80, where α- and κ-Tpm isoforms have the maximum differences in the amino acid sequences.

As during the MD calculations at high temperature, Tpm lost its stability in the C-terminal half of the molecules; we also took the first 16.5 nm-long part for the persistence lengths and bending stiffness analysis. At 300 K, ακTpm and κκTpm have similar persistence lengths estimated from 36 nm-long segments: ξ = 132 nm, *K*_b_ = 547 pN·nm^2^ (ακTpm), ξ = 132 nm, *K*_b_ = 545 pN·nm^2^ (κκTpm); ααTpm was slightly stiffer (Figure 6; Table 4). An increase in temperature to 323 K led to a decrease in bending stiffness. A visible difference is seen at high temperature where κκTpm becomes less stiff in the central region. The values estimated from the first halves of the graphs in Figure 6 are shown in Table 4. The results show that the most stable N-terminal part of Tpm becomes less rigid upon heating, and κκTpm is more thermosensitive than ααTpm and ακTpm. The difference between the isoforms is not surprising as they differ by the number of amino acid residues in the 2nd exon between the Tpm residues 39 and 77.

### 2.2. Interaction of the Tpm Isoforms with F-Actin

#### 2.2.1. Affinity of ααTpm and κκTpm to F-Actin

The affinity of Tpm to F-actin was determined by co-sedimentation followed by electrophoresis in SDS-PAGE. The affinity of ααTpm and κκTpm to F-actin did not differ, but ακTpm interacted much worse with F-actin: the concentration of tropomyosin required to achieve half-saturation was 1.23 ± 0.21 μM for ααTpm and 1.29 ± 0.21 μM for κκTpm. This value could not be obtained for ακTpm since, under these conditions, saturation is not achieved (Figure 7a).

#### 2.2.2. The Thermal Stability of the Tpm–F-Actin Complex

The thermal stability of the Tpm–F-actin complex was studied using the light scattering method. The thermal stability of the κκTpm complex with F-actin was lower than that of the ααTpm complex (Figure 7). The temperature of 50% dissociation of the complex with F-actin was 45.94 °C for ααTpm and 42.06 °C for κκTpm. The measurement error was no more than 0.02 °C. We were unable to estimate this parameter for ακTpm as it poorly binds F-actin.

### 2.3. Compering Regulatory Properties of ααTpm, κκTpm, and ακTpm

#### 2.3.1. Effects of ααTpm, κκTpm, and ακTpm on Calcium Regulation of Actin–Myosin Interaction

We compared the characteristics of the calcium dependence of the sliding velocity of thin filaments containing ααTpm, κκTpm, and ακTpm (pCa–velocity relationship) over pig atrial and ventricular myosin. Although ακTpm poorly binds F-actin, the calcium dependence of the sliding velocity of thin filaments containing it has a classical sigmoid character (Figure 8; Table 5). The characteristics of pCa–velocity relationship with ααTpm, κκTpm, and ακTpm did not differ with both atrial and ventricular pig cardiac myosins (Figure 8a,b; Table 5).

Previously, it was found that the calcium sensitivity of the tension of skinned muscle bundles from transgenic mice overexpressing κκTpm is lower than that of non-transgenic mice [7,9]. We compared the characteristics of the calcium dependence of the velocity of filaments containing κκTpm and ααTpm over rat ventricular myosin. With rat myosin, calcium sensitivity of thin filaments containing κκTpm was significantly lower than that with ααTpm (Figure 8; Table 5).

#### 2.3.2. Activation of Thin Filaments Containing ααTpm, κκTpm, and ακTpm by Myosin Cross-Bridges

In the calcium regulation of the actin–myosin interaction, an important role belongs to the mechanisms of cooperativity, one of which is the activation of myosin by cross-bridges (cross-bridge–cross-bridge cooperativity). We compared the activation of thin filaments containing ααTpm, κκTpm, and ακTpm by ventricular and atrial myosin. For this, we analyzed the dependence of the filament sliding velocity on the concentration of myosin loaded into the flow cell. It was found that activation of filaments containing κκTpm and ακTpm by atrial myosin requires less myosin than those containing ααTpm (Figure 9a, Table 6). The activation of thin filaments containing ααTpm, κκTpm, and ακTpm by pig ventricular myosin did not differ (Figure 9b, Table 6).

### 2.4. Effect of Tn on the Interaction of ααTpm, κκTpm, and ακTpm with F-Actin

It is known that the affinity of Tpm to F-actin is enhanced by TnT [19,20,21]. We hypothesized that Tn can interact differently with ααTpm, κκTpm, and ακTpm. We previously found that the troponin T1 fragment (TnT1; residues 1–158 for rabbit fast skeletal troponin T, P02641) containing 1 site interaction with Tpm [19] enhances the affinity of ααTpm to F-actin [22]. TnT1 decreased the Tpm concentration at which half of the actin was saturated, K_50%_, from 3 ± 0.2 µM to 1.25 ± 0.11 µM [22]. Here, we analyzed the effect of TnT1 on the affinity of κκTpm and ακTpm to F-actin. We found that TnT1 enhances the affinity of κκTpm (2.37 ± 0.21 µM for kkTpm, 0.88 ± 0.19 µM for kkTpm with TnT1) and ακTpm (0.71 ± 0.09 µM with TnT1) to F-actin (Figure 10a,b).

Here we found that ααTpm and κκTpm reduced the sliding velocity of F-actin (2.2 µm/s) by 50% and 30%, respectively, and ακTpm did not affect it. Adding TnT1 decreased the velocity of F-actin–Tpm filaments containing ααTpm and κκTpm but did not affect the sliding velocity of the filaments with ακTpm (Figure 10c). However, the addition of the whole Tn complex inhibited the sliding velocity of filaments containing ακTpm to a greater extent than ααTpm (Figure 10d).

## 3. Discussion

Tropomyosin and troponin are involved in the Ca^2+^ regulation of the myosin interaction with actin. Tropomyosin (Tpm) is a parallel α-helical coiled-coil dimeric protein ~40 nm long [3]. Tpm molecules bind each other in a head-to-tail manner forming a continuous strand that winds around the fibrous actin (F-actin) helix [3]. Tpm with troponin complex regulates the interaction of myosin with actin, thus controlling Ca^2+^ dependence of myocardial contraction. The structural and functional properties of Tpm determine the features of the cooperative binding of myosin to the actin filament and affect myocardial contraction. The human heart contains several isoforms of tropomyosin: alpha, beta, and kappa [3,4]. The main isoform of Tpm is alpha, while the remaining isoforms are minor components of the cardiomyocyte contractile apparatus. Here, for the first time, we analyzed in detail the structural and functional features of κκTpm.

Previously, it was shown with CD that κκTpm is less thermally stable than ααTpm [7]. Using DSC, we have analyzed the thermal unfolding and domain structure of ααTpm and κκTpm and confirmed that the thermal stability of the N-terminal part of the molecule is substantially lower in κκTpm than in ααTpm. These DSC data agree well with the results of MD simulations, which showed that the N-terminal part of Tpm becomes less rigid upon heating and κκTpm is more thermosensitive than ααTpm. Such reduced stability can result from the difference in the N-terminal exons of the TPM1 gene: exon 2a for κκTpm and exon 2b for ααTpm [4]. As exon 2 is located in the N-terminal part of the Tpm molecule, it is not surprising that its amino acid composition affects its thermal stability.

Based on structural modeling, Rajan et al. [7] suggested that, in vivo, the ακTpm heterodimers are more likely to form than the κκTpm homodimers. We found that mixing an equimolar amount of α- and κ-Tpm chains leads to the predominant formation of heterodimers (up to 90%; Figure 1). From these data, one can conclude that in the myocardium, kappa Tpm exists mainly in the form of a heterodimer with alpha Tpm. According to CD data, the ακTpm heterodimer stability is higher than that of ααTpm and κκTpm homodimers (Figure 2), which can explain the predominant heterodimer formation.

We studied the interaction of ααTpm, κκTpm, and ακTpm with F-actin by comparing the affinity of Tpm species to F-actin and the thermostability of the F-actin–Tpm complexes. We found that in contrast to the data of Rajan et al. [7], κκTpm and ααTpm have the same affinity to F-actin, while ακTpm interacts weakly with F-actin. The difference between the results of Rajan and ours can be explained by the difference in the ionic strength and pH of the solution of interacting proteins. It turned out that the F-actin–Tpm complex with κκTpm is less thermostable than that with ααTpm. The data obtained indicate that κκTpm interacts with F-actin weaker than ααTpm.

The lower interaction of κκTpm with F-actin resulted in a lesser inhibiting F-actin velocity in the in vitro motility assay than ααTpm; ακTpm did not affect the velocity of F-actin. The addition of the TnT1 fragment enhanced the interaction of all Tpm species with F-actin, and the K_50%_ values became close (Figure 10a,b). TnT1 suppressed the sliding velocity of F-actin–Tpm filaments containing ααTpm and κκTpm but did not affect the velocity of the filaments with ακTpm (Figure 10c,d). The formation of the overlap junction between the N- and C-ends of the Tpm molecule was shown to be important for the interaction of TnT1 with tropomyosin [23,24,25,26] and the Tpm regulatory function [14,21,22,27,28,29]. The ability of the N- and C-ends of ααTpm and κκTpm to interact was determined by measuring the viscosity of the solution. Since this measurement requires a high concentration of protein, we were unable to do this experiment with ακTpm. We found a stronger interaction between the ends of ααTpm molecules as compared with κκTpm molecules. The lack of influence of the TnT1 fragment on the velocity of F-actin–ακTpm filaments can be explained by assuming that the position of ακTpm on the actin filament in the presence of TnT1 differs from the κκTpm and ααTpm position. Previously, it was shown that the position of the tropomyosin strand on the actin filament depends on the isoform of Tpm [30,31]. The addition of the whole Tn complex inhibited the sliding velocity of filaments containing ακTpm even to a greater extent than ααTpm (Figure 10d). The whole Tn complex containing TnI more firmly anchors Tpm to F-actin.

We have studied the effect of ααTpm, κκTpm, and ακTpm on the calcium dependence of the sliding velocity of thin filaments over atrial and ventricular myosin in the in vitro motility assay. The characteristics of this dependence of Tpm isoforms were the same (Figure 8a,b, Table 5). Rajan et al. [6] showed that expression of human κTpm (72%) in transgenic mice decreased calcium sensitivity of trabeculae force. In several studies, it has been shown that cardiac myosin properties in large and small mammals differ [32,33,34]. Therefore, in addition to the pig myosin, we analyzed the effect of κκTpm on the calcium dependence of the thin filament velocity with myosin from the rat left ventricular myocardium. On the rat heart myosin, κκTpm decreased the calcium sensitivity of the filament sliding velocity and increased the maximal velocity (Figure 8c; Table 5), unlike that on the pig myosin. Thus, the effect of κTpm on the actin-myosin interaction and contractile function of the heart of large and small mammalians appears to be different.

We compared the activation of thin filaments comprising ααTpm, κκTpm, or ακTpm by ventricular and atrial myosin by the dependence of the filament velocity on the myosin concentration. We found that atrial myosin better activates thin filaments with κ-chain-containing Tpm than those with ααTpm (Figure 9; Table 6). As known, the myosin motor domain interacts not only with the actin of the thin filament but also with tropomyosin [35,36]. Alpha and beta isoforms of cardiac myosin heavy chain differ by the amino acid sequence of the motor domain [37], which can affect this interaction. In addition, light chains of atrial and ventricular myosin are different [35]; this also influences the actin-myosin interaction [36,37,38,39]. The atrial and ventricular myosin were shown to affect differently the activation of the thin filament containing Tpm with cardiomyopathic mutations and various isoforms of tropomyosin [40]. Using top-down mass spectrometry, Peng et al. [8] revealed that the level of κTpm expression in the human heart is significantly higher in atria than in ventricles. Thus, one can assume that κTpm is required for normal contraction of the atrial myocardium.

Peculiarities of the structural and functional properties of ααTpm and κκTpm are associated with the difference in 27 amino acid residues located between the 39th and 77th residues in the N-terminal part [7]. Most of the substitutions lead to destabilization of the dimer [7] and, consequently, to a lower stability of the κκTpm molecule compared to the ααTpm molecule. Part of these amino acid substitutions may affect the interaction of Tpm with F-actin and troponin complex, which we found experimentally.

## 4. Materials and Methods

The porcine heart was procured from a Kamensk-Uralsky public corporation. The rats used in the present study were treated according to Directive 2010/63/EU of the European Parliament. The experimental protocol was approved by The Animal Care and Use Committee of the Institute of Immunology and Physiology. Unless otherwise noted, all chemicals and reagents were purchased from Sigma-Aldrich (St. Louis, MO, USA).

Adult male Wistar rats at 10 weeks of age (250–300 g) were housed in the institutional vivarium with free access to food (Delta Feeds LbK 120 S-19, BioPro, Novosibirsk, Russian Federation) and water. Rats were deeply anesthetized with an intramuscular injection of 2% Xylazine (1 mL/kg body weight, Alfasan, Woerden, Netherlands) and Zoletil-100 (0.3 mL/kg body weight, Virbac, Carros, France), heparinized (5000 IU/kg, Ellara, Pokrov, Russian Federation), and euthanized by exsanguination. The heart was removed, and the free wall of the left ventricles was frozen in portions for the isolation of myosin and kept at −86 °C.

### 4.1. Proteins

All Tpm species in this work were recombinant human isoforms with an Ala-Ser N-terminal extension to imitate naturally occurring acetylation of native Tpm [41]. The ακ-Tpm heterodimer was obtained according to the method described by Kalyva et al. [42] with minor changes. Briefly, αα-homodimers containing N-terminal His-tag were mixed with κκ-homodimers, in a molar ratio of 1:1. This mixture was heated up to 65 °C for the Tpm dimers’ separation to monomeric chains and then cooled down to 25 °C and incubated at this temperature for dimer formation. This procedure led to the formation of a mix of αα-, κκ-, and ακ-dimers. The components of the mixture were separated by liquid chromatography on a His-TrapHP column. The dimers were separated by the imidazole gradient, and only ακ-heterodimers were collected. Finally, the His-Tag was removed by factor Xa protease (NEB, New England Biolabs, Hitchin, UK) at 4 °C overnight.

The cardiac Tn complex was extracted from the left ventricle of the pig [43]. Human adult cardiac troponin T (TNT3, or isoform 6 of P45379) fragment TnT1 was obtained as previously described [22]. TnT1 concentration was determined spectrophotometrically at 280 nm using an E^1%^ of 0.71 cm^−1^.

Myosin was extracted from the left ventricle and atrium of the porcine heart and the left ventricle of the rat hearts according to [44]. Atrial myosin from the porcine heart contained ~95% α-MHC, ~5% β-MHC, and atrial LCs; ventricular myosin contained 100% β-MHC and ventricular LCs. Rat ventricular myosin contained 90% α-MHC, 10% β-MHC, and ventricular LCs.

Skeletal rabbit muscle actin was prepared by standard methods [45]. F-actin was polymerized by the addition of 1 mM ATP, 4 mM MgCl_2_, and 100 mM KCl. When measuring the thermal stability of the Tpm-actin complex, F-actin was stabilized with a 1.5-fold molar excess of phalloidin. For the in vitro motility assay experiments, F-actin was labeled with a 2-fold molar excess of TRITC-phalloidin.

### 4.2. Differential Scanning Calorimetry (DSC)

The experiments were performed on a MicroCal VP-Capillary DSC differential scanning calorimeter (Malvern Instruments, Northampton, MA 01060, USA) at a heating rate of 1 °C/min in 30 mM Hepes-Na buffer (pH 7.3) containing 100 mM NaCl. The protein concentration was 2 mg/mL. In some experiments, the αTpm and κTpm samples were reduced before measurement by heating at 60 °C for 20 min in the presence of 3 mM DTT. After such a procedure, all Tpm samples were in an entirely reduced state [14,15,16]. In these experiments, the solution contained 3 mM DTT to prevent disulfide cross-linking between Cys residues in two chains of the Tpm molecule. In some other cases, the κTpm isoform was heated in the measuring cell of the calorimeter in the absence of reducing agents, especially to produce disulfide cross-linking between two chains of the κTpm molecule.

The reversibility of the heat sorption curves was assessed by reheating the sample immediately after it had cooled from the previous scan. The thermal unfolding of all Tpm species was fully reversible and thus can be considered to achieve thermodynamic equilibrium, thus making possible further deconvolution analysis of their DSC curves. The calorimetric traces were corrected for the instrumental background by subtracting a scan with the buffer in both cells. The temperature dependence of the excess heat capacity was further analyzed and plotted using Origin v. 7.5 software (MicroCal Inc., Northampton, MA, USA). The thermal stability of the proteins was described by the temperature of the maximum of the thermal transition (T_m_), and the calorimetric enthalpy (ΔH_ca_l) was calculated as the area under the excess heat capacity function. Deconvolution analysis of the heat sorption curves, i.e., their decomposition onto separate thermal transitions (calorimetric domains) by fitting the data to the non-two-state model [12], was as described previously [13,14,15,16].

### 4.3. Circular Dichroism

Far-UV circular dichroism (CD) spectra of the Tpm species at a concentration of 1.0 mg/mL were recorded at 5 °C on a Chirascan CD spectrometer (Applied Photophysics, Surrey, UK) in 0.02 cm cells [16]. Measurements of thermal unfolding were performed by following the molar ellipticity of Tpm at 222 nm over a temperature range of 5 to 70 °C at a constant heating rate of 1 °C/min. All measurements were performed in 20 mM Na phosphate, 100 mM NaCl, 1 mM MgCl_2_ 1 mM DTT buffer at pH 7.3.

### 4.4. The Affinity of Tpm to F-Actin

The affinity of Tpm species to F-actin without TnT1 and in the presence of TnT1 was estimated by a co-sedimentation assay as previously described [14,16,22]. The F-actin in a 10 μM concentration was mixed with TnT1 to the final concentration of 1.5 μM and Tpm in concentration increasing from 0 to 7.5 μM at 20 °C in 30 mM HEPES-Na buffer (pH 7.3) containing 255 mM NaCl and 200 mM NaCl without TnT1 to a final volume of 100 μL. After 40 min incubation, actin was pelleted with bound Tpm by ultracentrifugation at 100,000× *g* for 40 min. Equivalent samples of the pellet and the supernatant were subjected to SDS-PAGE. Protein bands were scanned and analyzed using ImageJ 1.53k software (Scion, Frederick, MD, USA).

### 4.5. Thermal Stability of the Tpm Complex with F-Actin

The stability of the Tpm–F-actin complex was measured by thermally induced changes in the light scattering [14,16,22]. The measurements were performed at 350 nm wavelength and a constant heating rate of 1 °C/min on a Cary Eclipse fluorescence spectrophotometer (Varian Australia Pty Ltd., Mulgrave, Australia) equipped with a temperature controller and thermoprobes. The solutions contained 20 μM F-actin and 10.5 μM Tpm in 30 mM HEPES-Na buffer (pH 7.3) with 100 mM NaCl. During the heating, Tpm dissociates from F-actin, and the light scattering intensity becomes equal to that of F-actin alone. The temperature dependence of the light scattering of F-actin alone was subtracted from experimental curves. The resulting curves were analyzed by fitting them to a sigmoidal decay function (Boltzman) using Origin v. 7.5 software (MicroCal Inc., Northampton, MA, USA). The main analyzed parameter was T_diss_, i.e., the temperature at which a 50% decrease in light scattering occurs.

### 4.6. Viscosity Measurement

We assessed the interaction of the N- and C-ends of Tpm by the viscosity measurement on a falling ball micro viscometer Anton Paar AMVn (Ashland, VA, USA) in a 0.5 mL capillary at 20 °C [22]. The specific Tpm solutions’ density was determined with an Anton Paar DMA 4500 density meter (Ashland, VA, USA) for accurate viscosity calculation. All measurements were performed at a Tpm concentration of 1.0 mg/mL (15 µM) in 30 mM HEPES-Na buffer (pH 7.3) containing 100 mM NaCl and 4 mM DTT. Each sample was measured threefold, and the results were averaged.

### 4.7. In Vitro Motility Assay

The in vitro motility assay experiments were done as described previously [16,22]. Myosin (300 µg/mL) in AB buffer (25 mM KCl, 25 mM imidazole, 4 mM MgCl_2_, 1 mM EGTA, and 20 mM DTT, pH 7.5) with 0.5 M KCl was loaded into an experimental flow cell with the nitrocellulose-coated inner surface. In 2 min, 0.5 mg/mL BSA was added for 1 min. Non-labeled F-actin in AB buffer with 2 mM ATP was added for 5 min. To form regulated thin filaments, 10 nM phalloidin-tetramethylrhodamine B isothiocyanate-labeled F-actin and 100 nM Tpm/Tn were added to the cell for 5 min. Unbound filaments were washed out with AB buffer. Finally, the cell was washed with AB buffer containing 0.5 mg/mL BSA, oxygen scavenger system (3.5 mg/mL glucose, 20 μg/mL catalase, and 0.15 mg/mL glucose oxidase), 20 mM DTT, 2 mM ATP, 0.5% methylcellulose, 100 nM Tpm/Tn, and appropriate Ca^2+^/EGTA in proportions calculated with Maxchelator program (http://www.stanford.edu/~cpatton/webmaxc/webmaxcS.html, accessed on 1 November 2018). In each flow cell, ten 30 s image sequences were recorded at 30 °C from different fields containing ~30–50 thin filaments. The filaments sliding velocities were measured using the GMimPro software (The Crick Institute, London, UK) [46].

To compare the regulatory properties of αTpm and κTpm, we analyzed the calcium dependence of the sliding velocity of thin filaments over atrial and ventricular myosin. Experiments with each Tpm isoform were repeated three times with de novo-prepared myosin. Means of individual experiments were fitted with the Hill equation: V = V_max_ × (1 + 10^n(pCa-pCa50)^)^−1^, where V and V_max_ are a velocity and the maximal velocity at saturating calcium concentration, respectively; pCa_50_ (i.e., calcium sensitivity) is pCa at which half-maximal velocity is achieved; and n is the Hill cooperativity coefficient. The parameters of individual experiments were averaged.

The effect of αTpm and κTpm on the cross-bridge, cross-bridge cooperativity was assessed by the dependence of the sliding velocity of thin filaments on the concentration of myosin added to the flow cell. The dependence of the sliding velocity on the myosin concentration was fitted with the Hill equation [47] V = V_max_ × c^h^ × (c_50_^h^ + c^h^)^−1^, where V_max_ is the maximal sliding velocity; c is myosin concentration; c_50_ is the concentration required to achieve half-maximal velocity; and h is the Hill coefficient.

Data analysis and statistical analysis were performed using Excel 16 (Microsoft Corp., Redmond, WA, USA) and Origin 8.0 (Origin Lab, Northampton, MA, USA).

### 4.8. MD Simulation

The MD simulation was performed using GROMACS v. 2019.3 [48]. The structure of a full αTpm molecule (*Sus scrofa*, PDB code 1C1G) [49] was used as a starting model. Several residue substitutions were made with the UCSF CHIMERA package [50,51] to build human WT Tpm1.1 (αTpm) and Tpm1.2 (κTpm) models. The model system was placed in a rectangular box extending 15 A in each direction from the protein filled with water molecules. Na^+^ and Cl^-^ ions were added to the system to ensure net zero charge and ionic strength of 0.15 M. The energy minimization, NVT and NPT equilibrations, and the MD simulation were carried out using the AMBER99SB-ILDN force field [52] and TIP3P water as previously described [16]. The duration of the MD runs was 200 ns, and the temperatures were set to 300, 315, and 323 K.

Snapshots of a 200 ns-long MD trajectory were taken every 200 ps. With the κκ-WT isoform at 300 K, two runs were performed, one recorded with a 20 ps time step and another with 200 ps, and their characteristics nearly coincided. Equilibration monitored by measuring the time course of the changes in RMSD from the initial energy-minimized structure was established within the first 5 ns. The algorithm for evaluating the persistence length, ξ, and bending stiffness, *K*_b_, of the Tpm coiled-coil was described previously [16]. In brief, the axis of the Tpm coiled-coil in each MD snapshot was approximated by a polygonal line using centroids of the Cα atoms of 11 consequent amino acids of each polypeptide chain (the 1st and 11th residues were taken with a weight factor of 0.5). The first segments of all lines were superimposed, and the time-average unit vectors t_0_(s) = <t(s)> of each segment were calculated (s is the distance along the polygonal line). Then, ξ and *K*_b_ were estimated using a generalization of the worm-like theory for a semi-rigid rod with a priori unknown intrinsically bent shape [53]: log<(t(s), t_0_(s))> = −s/ξ, *K*_b_ = ξk_B_T, where (,) denotes the scalar product of two vectors; <…> means time averaging; k_B_ is the Boltzmann constant; and T is the absolute temperature. As data points deviate from straight lines (Figure 6), the slope of the least square linear fit was used to evaluate ξ both for the N-terminal 16.5 nm fragment (residues 16–126) where the Tpm isoforms have differences in the amino acid sequence and for the 36 nm-long central part (residues 16–266). The terminal residues were excluded from the analysis as they were subjected to a larger disordering in the MD simulation. The occupancy of the backbone hydrogen bonds (*h*-bonds) was calculated with a Python script and the *h*-bond function of GROMACS. The average occupancy values for identical residues of both Tpm chains were plotted against the residue number to characterize the α-helix stability along the molecule.

## 5. Conclusions

We compared the structural features and functional properties of ααTpm, ακTpm, and κκTpm. We found that the formation of ακTpm heterodimers is thermodynamically favorable, and therefore, in the myocardium, κTpm most likely exists as ακTpm heterodimers. The thermal stability of the Tpm isoforms decreased in a row: ακTpm, ααTpm, κκTpm. The DSC results showed that the thermal stability of the N-terminal part of κκTpm is much lower than that of ααTpm. ακTpm almost did not interact with F-actin. The addition of the TnT1 fragment significantly increased the affinity of Tpm containing κ-chain to F-actin. Effects of κκTpm on the calcium regulation of actin–myosin interaction depended on the animal species and myosin isoform. With rat ventricular myosin, calcium sensitivity of thin filaments containing κκTpm was significantly lower than that with ααTpm, whereas with pig’s ventricular myosin, sensitivity did not differ. Atrial myosin better activated thin filaments containing κκTpm and ακTpm than filaments containing ααTpm. Based on these results and the fact that κTpm expression is higher in the atria, one can assume that the κ-chain of Tpm is more significant for the contractile function of the atrium than for the ventricle.

## Figures and Tables

**Figure 1 ijms-24-08340-f001:**
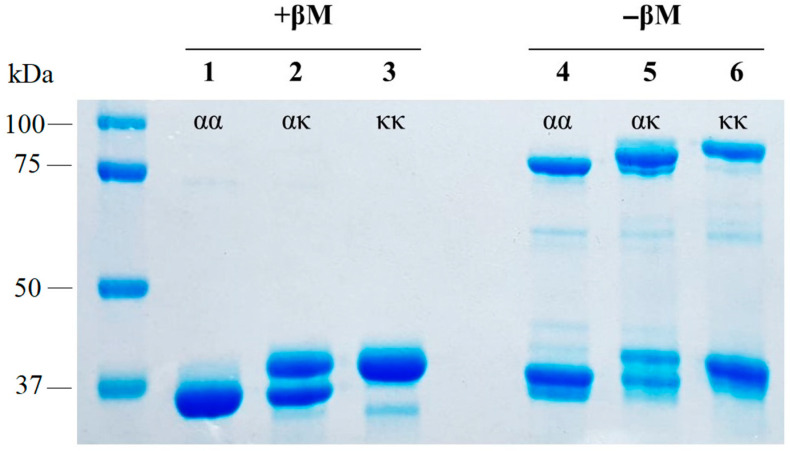
Gel electrophoresis of ααTpm, κκTpm, and their equimolar mixture after heating to 70 °C in the presence of β-mercaptoethanol (+βM; lanes 1–3) and without β-mercaptoethanol (−βM; lanes 4–6) with formation of -S-S- bonds.

**Figure 2 ijms-24-08340-f002:**
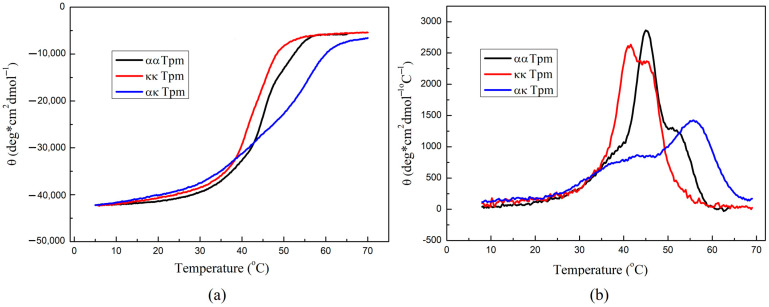
Circular dichroism measurements of thermal unfolding of ααTpm, ακTpm, and κκTpm. (**a**) Temperature dependencies of α-helix stability measured by ellipticity at 222 nm at a constant heating rate of 1 °C/min. (**b**) The first derivative profiles for the data are shown in (**a**). The protein concentration was 1.0 mg/mL.

**Figure 3 ijms-24-08340-f003:**
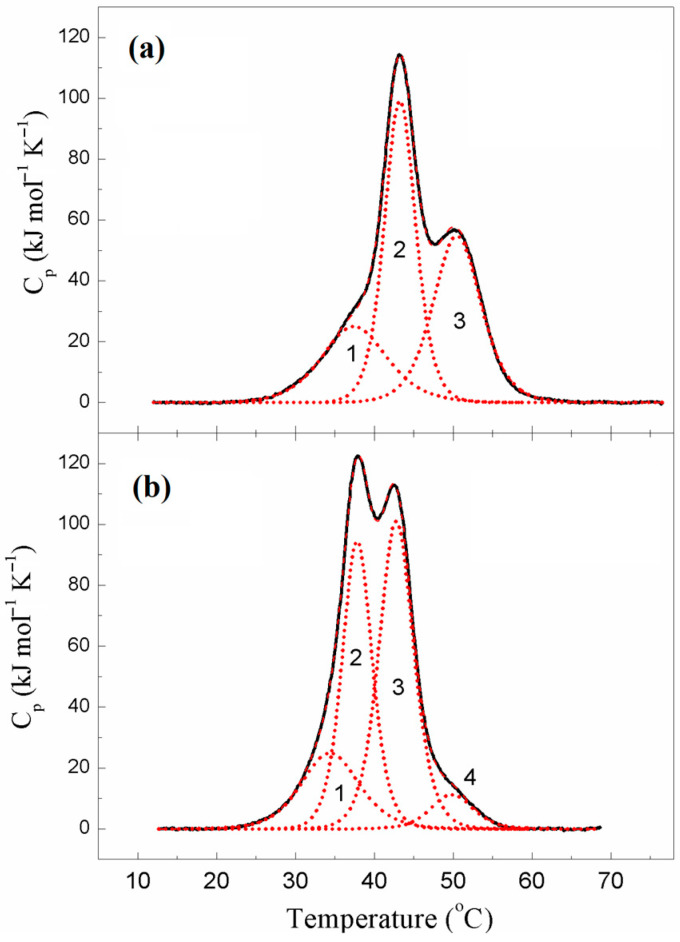
Temperature dependences of the excess heat capacity (Cp) monitored by DSC and deconvolution analysis of the heat sorption curves of ααTpm (**a**) and κκTpm (**b**) in a fully reduced state. The heating rate was 1 K/min. The curves were analyzed according to the non-two-state model [12]. Solid lines represent the experimental curves after the subtraction of instrumental and chemical baselines, and dotted lines represent the calorimetric domains obtained from fitting the data to the non-two-state model. 1 to 4 are domain numbers.

**Figure 4 ijms-24-08340-f004:**
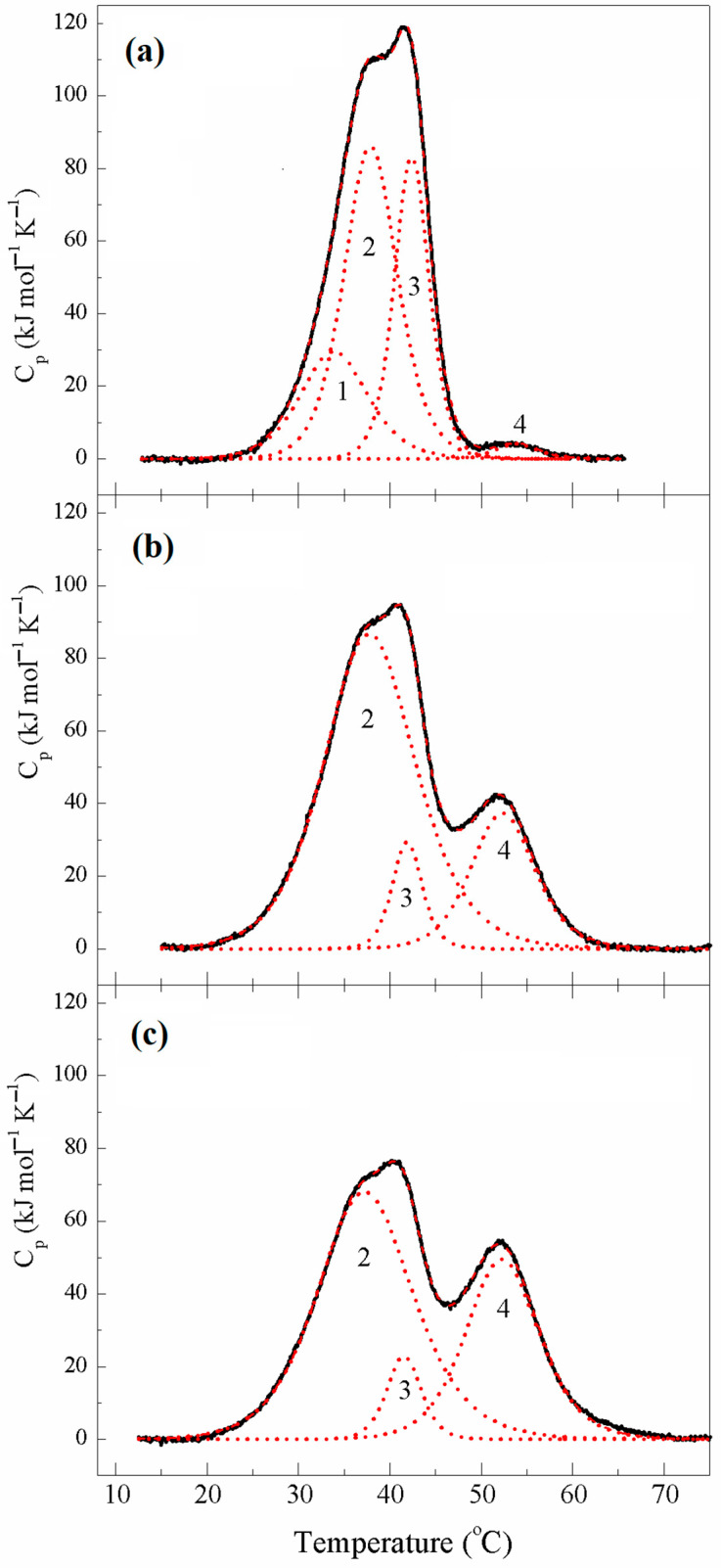
Temperature dependences of the excess heat capacity (C_p_) monitored by DSC and deconvolution analysis of the heat sorption curves of κκTpm obtained from 1st (**a**), 2nd (**b**), and 5th (**c**) heating of the sample in the absence of reducing agents. The heating rate was 1 K/min. The curves were analyzed according to the non-two-state model [12]. Solid lines represent the experimental curves after subtraction of instrumental and chemical baselines, and dotted lines represent calorimetric domains obtained from fitting the data with the non-two-state model. 1 to 4 are domain numbers.

**Figure 5 ijms-24-08340-f005:**
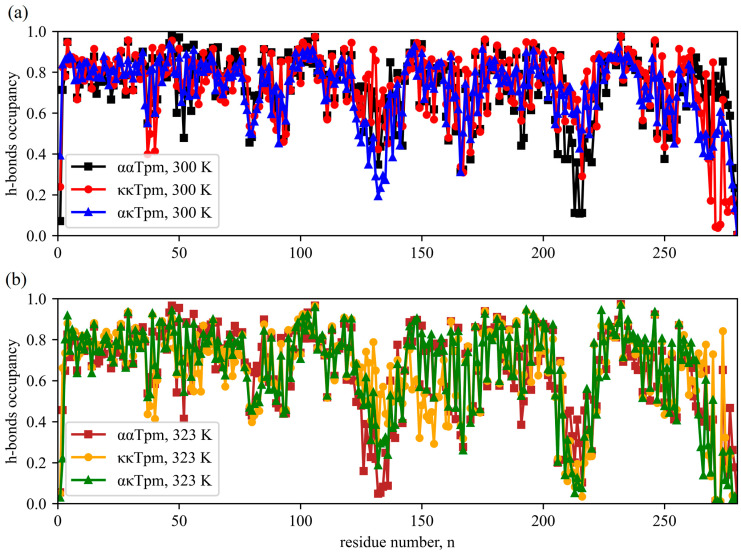
The time-averaged occupancies of the backbone *h*-bonds between the *n* and *n* + 4 residues in the molecular dynamics (MD) simulation of the ααTpm, κκTpm, and ακTpm at 300 K (**a**) and 323 K (**b**). The color code is in the inserted legend.

**Figure 6 ijms-24-08340-f006:**
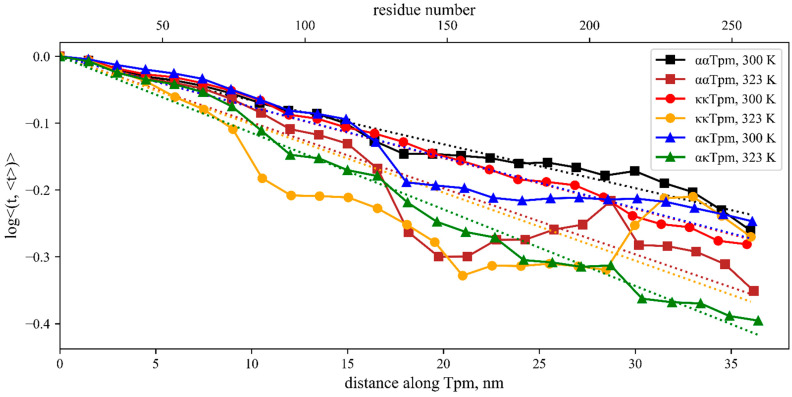
The persistence length of Tpm isoforms estimated from MD simulation of the ααTpm, κκTpm, and ακTpm at 300 K and 323 K as described in Methods. The color code is in the inserted legend. The least-square linear fits are shown by color dotted lines.

**Figure 7 ijms-24-08340-f007:**
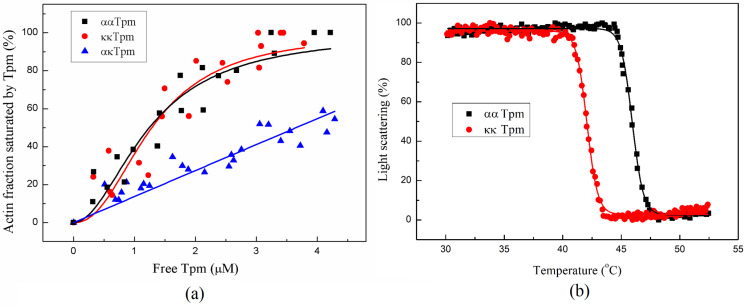
(**a**) Affinity of ααTpm, κκTpm, and ακTpm to F-actin. The fraction of F-actin bound to Tpm is plotted against the concentration of free Tpm found in the supernatant. (**b**) Normalized temperature dependencies of dissociation of the complexes of F-actin with ααTpm and κκTpm. A 100% value corresponds to the difference between the light scattering of the Tpm-F-actin complex measured at 25 °C and that of pure F-actin, which was temperature-independent within the temperature range used.

**Figure 8 ijms-24-08340-f008:**
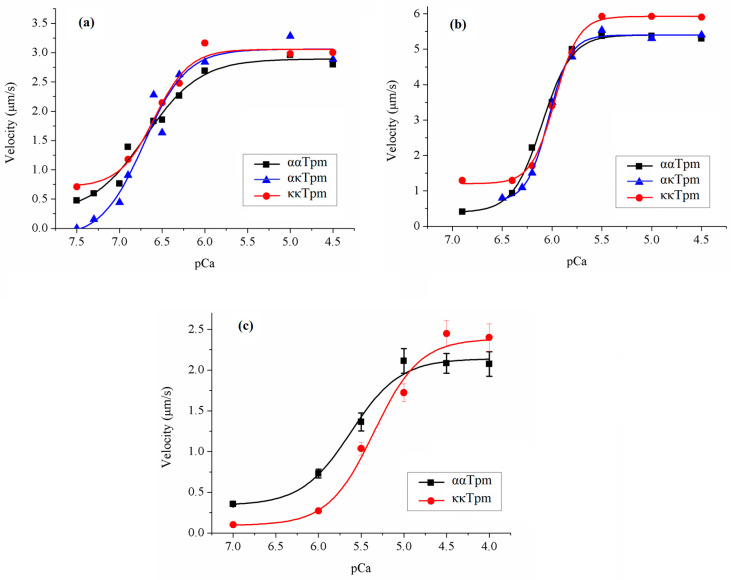
Calcium dependence of the sliding velocity of thin filaments containing ααTpm, κκTpm, and ακTpm over pig ventricular (**a**), pig atrial (**b**), and rat ventricular (**c**) myosin in the in vitro motility assay. Experiments were repeated three times. Each data point represents the mean ± the standard errors (S.D.) from three experiments. S.D. of experimental data in (**a**,**b**) are not shown in the figure; they did not exceed 10%. Experimental data were fitted with the Hill equation. Characteristics of the *p*Ca–velocity relationship are presented in Table 5.

**Figure 9 ijms-24-08340-f009:**
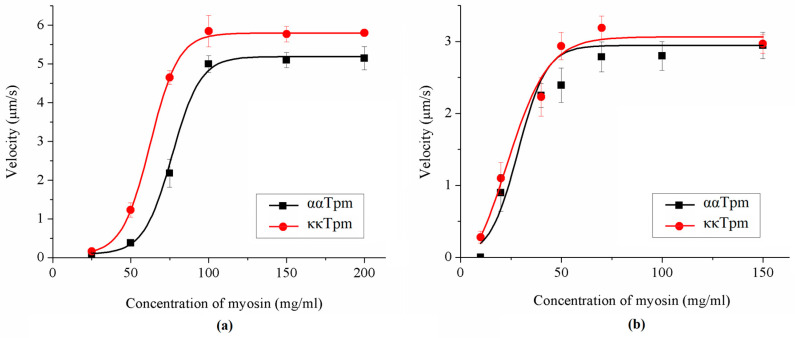
Example of the dependence of the sliding velocity of thin filaments containing ααTpm and κκTpm over atrial (**a**) and ventricular (**b**) myosin on its concentration in the in vitro motility assay. The sliding velocity dependence of ακTpm is completely identical to that of κκTpm. Experiments were repeated three times.

**Figure 10 ijms-24-08340-f010:**
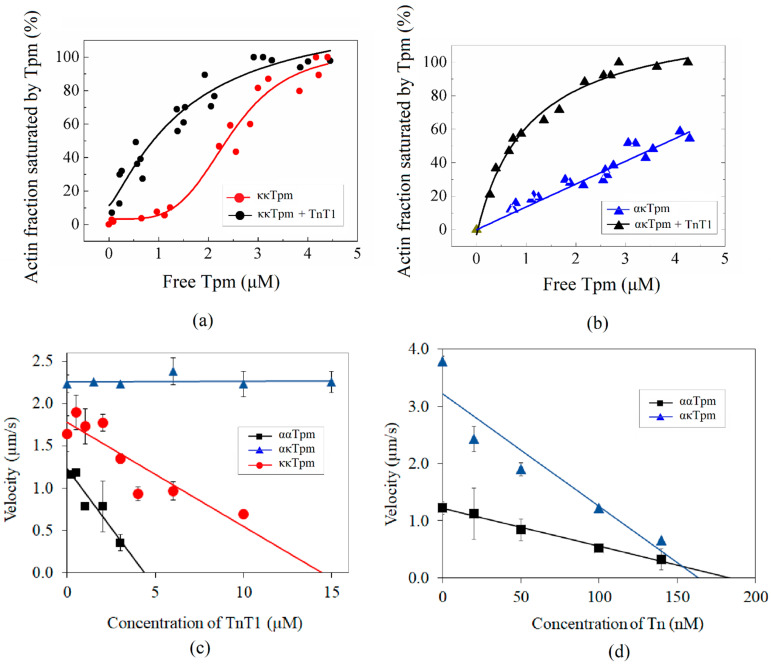
(**a**,**b**) Effect of the TnT1 fragment on the affinity of κκTpm (**a**) and ακTpm (**b**) to F-actin. (**c**,**d**) Dependence of the sliding velocity of F-actin–Tpm filaments over ventricular myosin on the concentration of the TnT1 fragment (**c**) and whole Tn complex (**d**) in the in vitro motility assay.

**Table 1 ijms-24-08340-t001:** Parameters of CD measurements of thermal unfolding.

Tpm	T_m_ (°C)		
	First Transition	Second Transition	Third Transition
ααTpm	36.6	45.2	52.2
ακTpm	36.8	44.6	55.8
κκTpm	41.4	45.6	

The error of the given values of transition temperature (T_m_) did not exceed ±0.8 °C.

**Table 2 ijms-24-08340-t002:** Calorimetric parameters obtained from the DSC data for calorimetric domains of ααTpm and κκTpm in the reduced state.

Tpm	T_m_ ^#^ (°C)	∆H_cal_ (kJ mol^−1^)	∆H_cal_ (% of Total)	Total ∆H_cal_ ^§^ (kJ mol^−1^)
**ααTpm**				1300
Domain 1	36.5	200	15	
Domain 2	43.2	580	45	
Domain 3	50.8	520	40	
**κκTpm (reduced)**				1400
Domain 1	34.4	250	18	
Domain 2	37.8	480	34	
Domain 3	42.8	590	42	
Domain 4	49.8	80	6	

The data were taken from Figure 3. ^#^ The error of the given values of transition temperature (T_m_) did not exceed ±0.2 °C. ^§^ The relative error of the given values of calorimetric enthalpy, ∆H_cal_, did not exceed ±10%.

**Table 3 ijms-24-08340-t003:** Calorimetric parameters obtained from the DSC data for calorimetric domains of κκTpm upon consecutive heatings of the sample in the absence of reducing agents.

κκTpm	T_m_ ^#^ (°C)	∆H_cal_ (kJ mol^−1^)	∆H_cal_ (% of Total)	Total ∆H_cal_ ^§^ (kJ mol^−1^)	Cross-Linking (%)
**1st heating**				1405	~5%
Domain 1	33.9	290	20		
Domain 2	37.9	670	48		
Domain 3	42.4	420	30		
Domain 4	53.9	25	2		
**2nd heating**				1625	~73%
Domain 2	37.9	1150	70		
Domain 3	41.9	130	8		
Domain 4	52.4	345	22		
**5th heating**				1580	~82%
Domain 2	37.4	945	60		
Domain 3	41.6	115	7		
Domain 4	52.3	525	33		

^#^ The error of the given values of transition temperature (T_m_) did not exceed ±0.2 °C. ^§^ The relative error of the given values of calorimetric enthalpy, ∆H_cal_, did not exceed ±10%.

**Table 4 ijms-24-08340-t004:** Persistence length (ξ) and bending stiffness (*K*_b_) of the 16.5 nm-long N-terminal part (first line in each cell) and the full-length Tpm (second line in each cell, bold). Values in brackets correspond to the 2nd half of the MD trajectory.

Tpm	300 K (27 °C)		315 K (42 °C)		323 K (50 °C)	
	ξ (nm)	*K*_b_ (pN·nm^2^)	ξ (nm)	*K*_b_ (pN·nm^2^)	ξ (nm)	*K*_b_ (pN·nm^2^)
ααTpm	147 (157) **152 (144)**	607 (650) **630 (597)**	116 (118) **106 (85)**	503 (512) **460 (369)**	115 (148) **101 (159)**	512 (662) **452(708)**
κκTpm	148 (171) **132 (129)**	614 (710) **545 (533)**	105 (107) **128 (129)**	458 (463) **559 (561)**	70 (93) **98 (118)**	312 (417) **437 (528)**
ακTpm	155 (162) **132 (184)**	640 (673) **547 (761)**	149 (161) **131 (169)**	648 (702) **569 (733)**	94 (121) **87 (110)**	421 (540) **389 (492)**

**Table 5 ijms-24-08340-t005:** Characteristics of pCa–velocity relationships for ventricular and atrial myosin.

Myosin	Tpm	V_max_ (µm/s)	V_0_ (µm/s)	pCa_50_
pig LV	ααTpm	2.9 ± 0.1	0.3 ± 0.2	6.69 ± 0.09
	κκTpm	3.1 ± 0.2	0.7 ± 0.2	6.59 ± 0.08
	ακTpm	3.1 ± 0.1	0 *	6.71 ± 0.09
pig LA	ααTpm	5.4 ± 0.1	0.4 ± 0.2	6.10 ± 0.02
	κκTpm	5.5 ± 0.1	1.2 ± 0.1	5.97 ± 0.01
	ακTpm	5.4 ± 0.1	0.7 ± 0.1	6.03 ± 0.01
rat LV	ααTpm	2.1 ± 0.1	0.3 ± 0.1	5.62 ± 0.08
	κκTpm	2.4 ± 0.1 *	0.2 ± 0.1	5.35 ± 0.09 *

LV, ventricular myosin; LA, atrial myosin; V_max_, the maximal sliding velocity of thin filaments at saturating Ca^2+^ concentration; V_0_, the velocity at low Ca^2+^ concentrations; pCa_50_, pCa at which V is half-maximal. Experiments were repeated three times. Parameters of the Hill equation are represented as mean ± S.D. The symbol * denotes differences in characteristics of pCa–velocity relationships for thin filaments containing κκTpm and ακTpm from that with ααTpm (*p* < 0.05).

**Table 6 ijms-24-08340-t006:** Characteristics of the dependence of filament velocity on myosin concentration.

Myosin	Tpm	V_max_ (µm/s)	C_50_ (µg/mL)
LA	ααTpm	5.2 ± 0.1	76.5 ± 2.1
	κκTpm	5.8 ± 0.01	62.5 ± 0.4 *
	αĸTpm	5.4 ± 0.2	55.1 ± 1.0 *
LV	ααTpm	2.8 ± 0.2	30.1 ± 1.0
	κκTpm	3.0 ± 0.1	23.4 ± 5.2
	αĸTpm	2.9 ± 0.3	30.2 ± 5.1

LV—ventricular myosin; LA—atrial myosin. Experiments were repeated three times. Characteristics of the dependence of filament velocity on myosin concentration are represented as mean ± S.D. The symbol * denotes differences in C_50_ for thin filaments containing κκTpm and ακTpm from that with ααTpm (*p* < 0.05).

## Data Availability

The datasets generated for this study are available on request to the corresponding author.

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
