# Peer review of "Structural and Functional Properties of Kappa Tropomyosin"

_ijms, 2023, doi:10.3390/ijms24098340_

Round 1

Reviewer 1 Report

Kopylova GV and co-workers in the article "Structural and Functional Properties of kappa Tropomyosin" studied the role of kappa tropomyosin (κTpm) in the regulation of actin-myosin interactions in the myocardium. They analyzed the structural features of κTpm using circular dichroism (CD), differential scanning calorimetry (DSC), viscosity measurements, molecular dynamics simulations and interactions of Tpm with actin, myosin and Tn using cosedimentation assay, light scattering and in vitro mobility assay. In addition, the authors compared the properties of κTpm with the αTpm homodimer and the ακ heterodimer.

As research on these proteins and actin multiple functions is intensive, in my opinion it is useful paper for the scientific community to extend the knowledge on the subject. However, the manuscript still needs some major and minor corrections:

1.     The first sentence in the Abstract should be divided into two separate ones.

2.     βTpm is Tpm2.2 not Tpm2.1 (line 44). Tpm2.1 is expressed in smooth muscle.

3.     I do not think one can say that κTpm is a minor protein (line 69). Did the authors mean that κTpm is present in the cardiac muscle in small amounts?

4.     I think it would be helpful for readers to show in Introduction or in Discussion comparison between exon 2a and 2b.

5.     Can the authors explain how they calculated that after mixing αα and κκ chains in equimolar amounts 90% of heterodimers was obtained  (Figure 1)? The bands are a bit blurry on the gel and I cannot see it is 90%. Was the procedure repeated and each time it came out the same? If this has only been done once, it should be repeated.

6.     In the same figure, the molecular weights for the standard lane are missing. In the caption to the figure, the authors wrote that the -S-S bonds are formed in the presence of β-mercaptoethanol and without β-mercaptoethanol... It does not make sense.

7.     Was actin stabilized with phalloidin for all experiments (lines 450-451)? If yes, why?

8.     I find no justification for the salt conditions used in some experiments. Why did the authors use 255 mM NaCl in the cosedimentation test and 0.5 M KCl in the in vitro mobility test, while in other experiments they used salt with a standard concentration of 0.1 M? Can the authors explain why? In addition, the above conditions used in the actin binding assay may have resulted in the lack of saturation of actin filaments with tropomyosin heterodimer. Perhaps the authors should repeat the experiment using a salt concentration of 0.1-0.15 M? This may improve affinity.

9.     My biggest doubts concern the cosedimentation assay results:

a)     The experimental points in Figure 7a are rather scattered. I wonder if there could have been an isoform swap (between α and κ) in any of the experiments? For this reason, more data is needed to obtain reliable results.

Perhaps this is the reason for the difference between the data obtained in this work compared to the work of Rajan S et al. 2010.

b)     Do I understand correctly that only heterodimers were cross-linked with DTNB? If so, the authors should check whether and how cross-linking with this compound affects the binding of Tpm to actin using appropriate controls. Otherwise, it is probably impossible to compare non-crosslinked homodimers with a crosslinked heterodimer in one graph. I think crosslinking  decreases binding abbilities of Tpm.

c)     In my opinion, in Figure 10a, b, the reader should see partly the same as in Figure 7a (curves showing binding in the absence of Tn), but this is not the case. I am completely lost. For example, the affinity of κκTpm for actin in Figure 10a is 2.37 µM, while in Figure 7a it is 1.29 µM. That is almost a double difference. Was the method described inaccurately or did I miss something? Or maybe it is worth going back to raw data?

d)     In Fig. 10b in the legend should be ακTpm+TnT1 and triangles instead of dots for ακTpm on the graph.

e)     In Fig In Fig. 10a, 100 on the Y-axis and 5 on the X-axis are missing.

f)      There seems to be some inconsistency regarding the result of heterodimer formation and DSC measurements. The authors wrote that DSC measurements require a high concentration of Tpm (2mg/ml) and therefore analyses of heterodimers were not possible. Does this mean that the heterodimer production yield was low? If so, it cannot be said that the formation of ακTpm is thermodynamically favorable and the solution is dominated by heterodimers (results described in section 2.1.1).

10.  Why are there no data for ακTpm when measured in the presence of myosin ratLV (Table 5)?

11.  Nomenclature (e.g. use only κTpm, not Tpm1.2 like in 2.1.3 paragraph title), units on graphs (e.g. Figure 10 c,d), and font size should be standardized.

12.  Please, check the typos e.g. in References 27 should be: Colpan and Tolkatchev.

13.  No. 54 it is not a reference.

Author Response

Response to reviewers’ comments

We are grateful to both Reviewers for their valuable and useful comments on our manuscript. We have corrected the manuscript according to their suggestions. Our point-to-point responses to the comments are listed below. Changes were made in the manuscript to meet the Reviewers' suggestions (highlighted yellow).

Reviewer 1

  1. The first sentence in the Abstract should be divided into two separate ones.

Corrected as suggested.

  1. βTpm is Tpm2.2 not Tpm2.1 (line 44). Tpm2.1 is expressed in smooth muscle.

We apologize for this misprint. Corrected.

  1. I do not think one can say that κTpm is a minor protein (line 69). Did the authors mean that κTpm is present in the cardiac muscle in small amounts?

Yes, we meant small amounts. Corrected as suggested.

  1. I think it would be helpful for readers to show in Introduction or in Discussion comparison between exon 2a and 2b.

Thank you for this advice. A comparison of the sequences of these exons has already been done by Geeves et al. (J. Muscle Res. Cell Motil., 2015). We added reference (5) to that work.

  1. Can the authors explain how they calculated that after mixing αα and κκ chains in equimolar amounts 90% of heterodimers was obtained (Figure 1)? The bands are a bit blurry on the gel and I cannot see it is 90%. Was the procedure repeated and each time it came out the same? If this has only been done once, it should be repeated.

The procedure was carried out repeatedly throughout the study to control heterodimer formation for other experiments. We apologize for the misleading in the applied MS. We measured the density of the bands only in the gel presented in Figure 1 with the ImageJ program. The densities in other gels were estimated by eye and the results each time were about the same. The sentence describing the content of the heterodimer is now corrected.

  1. In the same figure, the molecular weights for the standard lane are missing. In the caption to the figure, the authors wrote that the -S-S bonds are formed in the presence of β-mercaptoethanol and without β-mercaptoethanol. It does not make sense.

We agree with Reviewer’s remarks. Necessary corrections to the MS and Figure 1 were made.

  1. Was actin stabilized with phalloidin for all experiments (lines 450-451)? If yes, why?

No, it was not the case for all experiments. Phalloidin was used in the experiments for studying the complex stability. For the in vitro motility assay experiments, F-actin was labeled with TRITC-phalloidin. Appropriate changes to the MS were made.

  1. I find no justification for the salt conditions used in some experiments. Why did the authors use 255 mM NaCl in the cosedimentation test and 0.5 M KCl in the in vitro mobility test, while in other experiments they used salt with a standard concentration of 0.1 M? Can the authors explain why? In addition, the above conditions used in the actin binding assay may have resulted in the lack of saturation of actin filaments with tropomyosin heterodimer. Perhaps the authors should repeat the experiment using a salt concentration of 0.1-0.15 M? This may improve affinity.

In most experiments, we used 0.1M NaCl to decorate the actin filament with tropomyosin. In the in vitro motility assay, ionic strength was about the same. A 0.5 M KCl was used only at the stage of assembling the experimental cell to prevent the myosin molecules from sticking before they interact with the nitrocellulose-coated cell surface.

A 200 mM NaCl was used only in the co-sedimentation assay, where we evaluated the affinity of tropomyosin to the actin filament. The influence of the TnT1 fragment on the interaction of tropomyosin with the F-actin was estimated at 255 mM NaCl.  Now it is detailed in the Materials and Methods section.

The high ionic strength in the co-sedimentation assay was used not accidentally. At this condition, actin saturation with Tpm isoforms achieves not instantly, which makes it possible to record saturation curves at a convenient concentration range for analysis with SDS-PAGE. The addition of the TnT1 fragment increases the Tpm affinity to actin; that is why we raised the ionic strength to 255 mM. Of course, the experiment can be performed at 100–150 mM NaCl. This would increase the Tpm affinity to the actin filament, but the saturation by heterodimers not necessarily be achieved, and so the error in the K50% for homodimers would greatly increase.

  1. My biggest doubts concern the cosedimentation assay results:
  2. a) The experimental points in Figure 7a are rather scattered. I wonder if there could have been an isoform swap (between α and κ) in any of the experiments? For this reason, more data is needed to obtain reliable results.

Swapping the tropomyosin dimers in the gel is impossible due to the difference in their electrophoretic mobility.

The co-sedimentation assay cannot be attributed to the precise methods. High data scattering is its unavoidable feature (e.g., Mirza et al., J Biol Chem.  2007; Gupte et al., J Biol Chem. 2014; Barua et al., J Biol Chem. 2013). This method is still used widely for studying the affinity of polymer molecules. It is considered, that affinities are different if the Tpm concentrations required to achieve half-saturation are two-fold distinct at least.

Perhaps this is the reason for the difference between the data obtained in this work compared to the work of Rajan S et al. 2010.

In Fig. 1, an example of a co-sedimentation assay of ααTpm and κκTpm binding to F-actin is shown, which demonstrates that κκTpm interacts well with F-actin. We do not have any explanation for the difference in the results. Many possible reasons may affect the results of different works. The data may depend on the experimental conditions: quality of isolated actin, length of the actin filaments, etc.

Fig. 1. Example of co-sedimentation assay of ααTpm and κκTpm binding to F-actin.

  1. b) Do I understand correctly that only heterodimers were cross-linked with DTNB? If so, the authors should check whether and how cross-linking with this compound affects the binding of Tpm to actin using appropriate controls. Otherwise, it is probably impossible to compare non-crosslinked homodimers with a crosslinked heterodimer in one graph. I think crosslinking decreases binding abbilities of Tpm.

Tropomyosin dimers were crosslinked only to check the heterodimer formation and to estimate the fraction of the heterodimer in the equimolar mixture. In all other experiments, all molecules were in reduced conditions. Chromatography on HisTrap HP columns made it possible to isolate the pure fraction of the heterodimer in the reduced form.

  1. c) In my opinion, in Figure 10a, b, the reader should see partly the same as in Figure 7a (curves showing binding in the absence of Tn), but this is not the case. I am completely lost. For example, the affinity of κκTpm for actin in Figure 10a is 2.37 µM, while in Figure 7a it is 1.29 µM. That is almost a double difference. Was the method described inaccurately or did I miss something? Or maybe it is worth going back to raw data?

It is our, not your, fault and we apologize for that. It is now corrected in section 4.4. The ionic strength in experiments with and without the TnT1 fragment was different. In the experiments where we evaluated the affinity of tropomyosin for the actin filament (Figure 7a), it was 200 mM. In the experiments with the TnT1 fragment (Figure 10), the ionic strength for all preparations was 255 mM. This explains the two-fold difference in the constant.

  1. d) In Fig. 10b in the legend should be ακTpm+TnT1 and triangles instead of dots for ακTpm on the graph.

It is corrected.

  1. e) In Fig. 10a, 100 on the Y-axis and 5 on the X-axis are missing.

It is corrected.

  1. f) There seems to be some inconsistency regarding the result of heterodimer formation and DSC measurements. The authors wrote that DSC measurements require a high concentration of Tpm (2mg/ml) and therefore analyses of heterodimers were not possible. Does this mean that the heterodimer production yield was low? If so, it cannot be said that the formation of ακTpm is thermodynamically favorable and the solution is dominated by heterodimers (results described in section 2.1.1).

We only meant that the DSC method consumes a lot of protein. As far as this sentence could be interpreted ambiguously, we removed it. The point is that for measuremets of the heat absorption of heterodimers, we used CD instead of DSC since deconvolution of the heat absorption curves obtained with DSC cannot be made with the heterodimers. The second heating for such preparations would not coincide with the first one as heterodimers destroy during the first heating, the chains dissociate one from another, and an admixture of other forms appears at the subsequent heatings. As compared to CD, DSC consumes significantly more protein.

  1. Why are there no data for ακTpm when measured in the presence of myosin rat LV (Table 5)?

To study the effect of κTpm on calcium regulation of heart contraction, Rajan et al. (2010) used a Tg mouse model overexpressing κTpm up to 90% of the total Tpm. In this case, κTpm was predominantly a homodimer. Therefore, with rat myosin, we used only κκTpm.

  1. Nomenclature (e.g. use only κTpm, not Tpm1.2 like in 2.1.3 paragraph title), units on graphs (e.g. Figure 10 c,d), and font size should be standardized.

It is corrected.

  1. Please, check the typos e.g. in References 27 should be: Colpan and Tolkatchev.

It is corrected.

  1. No. 54 it is not a reference.

It is corrected.

Reviewer 2 Report

The paper performs a deep biochemical study on the formation of heteromers in myocardium, and concludes that from the thermodynamic point of view the most likely form is the formation of ακTpm heterodimers. The reserch is well designed and the conclusions are based on the experimental data presented so the study is very ellegant and well written.

Nevertheless I have found that some important information is missing in some figures or in the materials and methods.

Figures 5 and 6: are provided without error bars. Each point represents a single measure? If this is the case, how many times was the experiment repeated? And results were reproducible?

Table 5 and 6 and figure 8 and 9: Here we have error bars, but the question is, how many replicas per point? I miss the "n" number in the figure legend. Please include this essential information. 

Which software was used for the statistics and SD calculation? Please include this information in mat and met.

Author Response

Response to reviewers’ comments

We are grateful to both Reviewers for their valuable and useful comments on our manuscript. We have corrected the manuscript according to their suggestions. Our point-to-point responses to the comments are listed below. Changes were made in the manuscript to meet the Reviewers' suggestions (highlighted yellow).

Reviewer 2

Figures 5 and 6: are provided without error bars. Each point represents a single measure? If this is the case, how many times was the experiment repeated? And results were reproducible?

In Figures 5 and 6 each point represents an average value over 1000 time points in a single 200 ns-long MD run. One run (kkTpm at 300 K) was repeated with different randomly selected initial conditions and trajectory snapshots were recorded with a shorter time step of 20 ps. The results obtained were very similar to those in Figures 5 and 6. This is mentioned in the Methods section.

Table 5 and 6 and figure 8 and 9: Here we have error bars, but the question is, how many replicas per point? I miss the "n" number in the figure legend. Please include this essential information. 

It is corrected.

Which software was used for the statistics and SD calculation? Please include this information in mat and met.

It is corrected.

Round 2

Reviewer 1 Report

I consider the manuscript improved. The authors answered all questions, that is why I believe the manuscript is suitable for publication.